# Sequential sensory and decision processing in posterior parietal cortex

**Guilhem Ibos[1]\*, David J Freedman[1,2]**

[1]Department of Neurobiology, The University of Chicago, Chicago, United States; [2]Grossman Institute for Neuroscience, Quantitative Biology and Human Behavior, The University of Chicago, Chicago, United States

**Abstract** Decisions about the behavioral significance of sensory stimuli often require comparing sensory inference of what we are looking at to internal models of what we are looking for. Here, we test how neuronal selectivity for visual features is transformed into decision-related signals in posterior parietal cortex (area LIP). Monkeys performed a visual matching task that required them to detect target stimuli composed of conjunctions of color and motion-direction. Neuronal recordings from area LIP revealed two main findings. First, the sequential processing of visual features and the selection of target-stimuli suggest that LIP is involved in transforming sensory information into decision-related signals. Second, the patterns of color and motion selectivity and their impact on decision-related encoding suggest that LIP plays a role in detecting target stimuli by comparing bottom-up sensory inputs (what the monkeys were looking at) and top-down cognitive encoding inputs (what the monkeys were looking for).

## Introduction

Detecting behaviorally important stimuli in a visually cluttered environment (such as a predator in a field of high grass) is a difficult, yet vital ability. Solving such tasks relies on our ability to voluntarily select and enhance the representation of behaviorally relevant spatial and non-spatial features. A large corpus of studies has shown that both space-based attention (SBA) and feature-based attention (FBA) increase the activity of cortical visual neurons selective to the relevant features and positions (e.g. from V4 [*Reynolds et al., 2000*; *Connor et al., 1997*; *McAdams and Maunsell, 2000*] or MT [*Martinez-Trujillo and Treue, 2004*; *Seidemann and Newsome, 1999*]). Recently, we proposed that these attention-dependent gain modulations strengthen the bottom-up flow of relevant information and facilitate their integration and representation by downstream areas (*Ibos and Freedman, 2014*, *2016*). Specifically, we showed that both SBA and FBA modulations of lateral intraparietal area (LIP) neurons were consistent with LIP linearly integrating inputs from attention-modulated activity in upstream visual cortical areas.

Our previous studies and the model framework which accompanied them (*Ibos and Freedman, 2014*, *2016*) can potentially account for the encoding of individual spatial and non-spatial features in LIP. However, it did not address the role of LIP in solving tasks in which decisions rely on grouping different sensory feature representations and comparing them to an internal cognitive model of task-relevant information. Why does LIP integrate the flow of bottom-up sensory information? Why does LIP represent sensory information that is already reliably encoded in upstream areas? In the past 15 years, a large number of hypotheses about the role of LIP have emerged in the literature. LIP has been proposed to encode stimuli behavioral salience (*Ipata et al., 2009*; *Bisley and Goldberg, 2003*, *2010*; *Arcizet et al., 2011*; *Gottlieb et al., 1998*; *Leathers and Olson, 2012*), to transform sensory evidence into decisions about the target of saccadic eye movements (*Leon and Shadlen, 1998*; *Shadlen and Newsome, 2001*; *Gold and Shadlen, 2007*; *Huk and Shadlen, 2005*)

\*For correspondence:
guilhemibos@gmail.com

**Competing interests:** The authors declare that no competing interests exist.

or to encode cognitive signals such as rules (*Stoet and Snyder, 2004*) or abstract categories (*Freedman and Assad, 2006*; *Swaminathan and Freedman, 2012*; *Sarma et al., 2016*) independently of LIP's role in spatial processing (*Rishel et al., 2013*). The present study focuses on understanding how LIP jointly encodes sensory, cognitive and decision-related information during a complex memory-based visual-discrimination task in which decisions rely on comparing the identity of observed stimuli to the identity of a remembered stimulus.

We trained two monkeys to perform a delayed-conjunction matching (DCM) task. At the beginning of each trial, monkeys were cued by one of two sample stimuli about the identity (conjunctions of color and motion-direction features) and position of the upcoming target stimulus. After a delay, successive visual test stimuli were presented simultaneously at two positions. Monkeys were required to identify test stimuli matching the position, color and motion-direction of the sample by releasing a manual lever to receive a reward. This task design allowed us to compare how LIP neurons encode the identity of the test stimuli as well as their match status.

We show that LIP neurons jointly encode the identity and match/non-match status of test stimuli. Specifically, LIP contains a continuum of mixed selectivity from purely identity-selective neurons to purely match-selective neurons. Interestingly, the identity of the stimuli was encoded prior to their match-status. Moreover, the dynamics of identity-selective signals were not predictive of monkeys' behavioral reaction times (RT, i.e. timing between test stimulus onset and manual response), while match-selective signals and RT were correlated. Furthermore, this relationship appeared to be independent of motor-preparation signals, suggesting that match-selectivity reflected decision-related process. Finally, we tested how neurons combine independent signals relative to the color and the direction of test stimuli in order to encode their conjunction. We show that identity-selective neurons combine color and direction signals linearly. However, the representation of relevant conjunctions by match-selective neurons was super-additive. Together, these results suggest that, along its visuo-decision continuum, LIP integrates, combines and transforms sensory information into decisions about the relevance of the test stimuli. Such a signal could alert monkeys about the presence of behaviorally relevant stimuli.

## Results

The goal of this study was to characterize how LIP encodes both the visual features and behavioral significance of visual stimuli. These data have been partially presented in two previous reports in which we studied the effects of SBA and FBA on LIP spatial and non-spatial selectivity (*Ibos and Freedman, 2014*, *2016*).

### Task and behavior

We trained two macaque monkeys to perform two versions of the DCM task (two-location DCM *Figure 1* and one-location DCM *Figure 1—figure supplement 1*) presented in two previous reports (*Ibos and Freedman, 2014*, *2016*). Trials were initiated when monkeys held a manual lever and fixated a central fixation point. In the two-location DCM task, a sample stimulus then appeared either inside or outside the receptive field (RF) of the neuron being recorded (450 ms). After a delay (450 ms), a succession of one to four test stimuli (450 ms each) was presented at the location of the sample stimulus. In the one-location DCM, the sample (550 ms), always presented inside LIP neurons' RFs, was followed by a delay (550 ms) and a succession of one to three test stimuli (550 ms each). In the two-location DCM task only, distractor stimuli were simultaneously presented at the opposite location (180° in the opposite hemifield). In both versions of the DCM task, test and distractor stimuli were conjunctions of one of eight motion-directions and one of eight colors along a continuum from yellow to red (*Figure 1B*). The sample stimulus was either composed of yellow dots moving in a downward direction (sample A) or of red dots moving in an upward direction (sample B). The monkeys' task was to release the lever when any of the test stimuli matched the position, color and motion-direction of the sample stimulus. On 20% of trials (25% in the one-location DCM task), no target stimulus was presented (catch trials) and monkeys had to withhold their response until the end of the last test stimulus to get a reward. In the two-location DCM task, test stimuli (presented at the sample location) were randomly picked among three types of stimuli: (1) target stimuli, matching the sample, (2) opposite-target stimuli (i.e. the sample stimulus which was not presented during that trial, e.g. test stimulus A during sample B trials), (3) any of the remaining conjunctions of color and

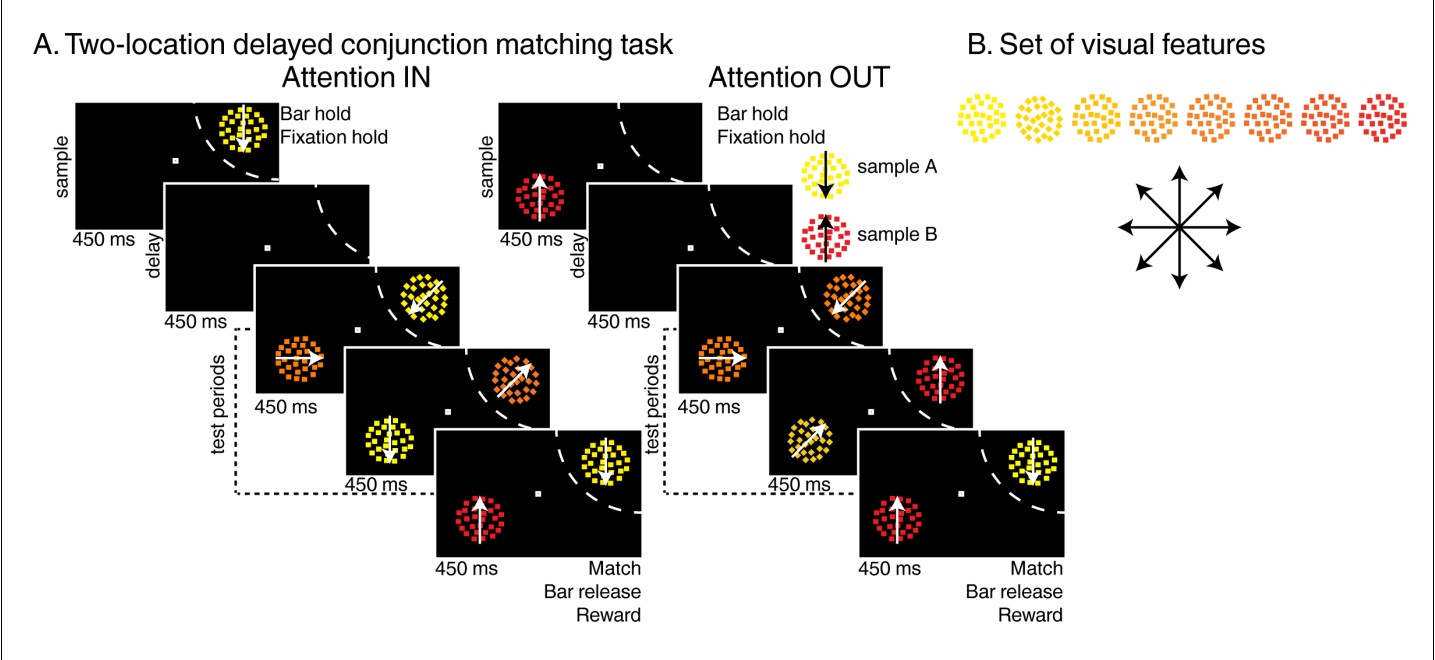

**Figure 1.** Task: (**A**) Two-locations delayed conjunction matching task. One of two sample stimuli was presented for 450 ms. The sample could either be sample A (yellow dots moving downwards) or sample B (red dots moving upwards). After a delay of 450 ms, one to four test stimuli were successively presented at the sample position in succession for 450 ms each while as many distractors were simultaneously presented in the opposite hemifield. In the attention IN condition, sample and test stimuli were presented in the receptive field (RF) of the recorded neuron (dashed arc, not shown to monkeys). In the attention OUT condition, sample and test stimuli were presented outside the RF while distractors were located inside the RF. All stimuli were conjunctions of one color and one direction. To receive a reward, monkeys had to release a manual lever when the test stimulus matched the sample in both color and direction and to ignore the distractors. On 20% of trials, none of the test stimuli matched the sample, and monkeys had to hold fixation and withhold their manual response to receive a reward. (**B**) Stimulus features: eight colors and eight directions were used to generate 64 different test/distractor stimuli. Colors varied from yellow to red, and directions were evenly spaced across 360 degrees.

The following figure supplement is available for figure 1:

**Figure supplement 1.** Task: (**A**) One location delayed conjunction matching task (*Ibos and Freedman, 2014*).

motion-direction. Distractor stimuli (presented at the location opposite the sample) were pseudo-randomly picked among the entire set of 64 stimuli. Correct behavioral responses were rewarded with a drop of juice. Distractor stimuli were always behaviorally irrelevant and had to be ignored. In these tasks, the behavioral relevance of test stimuli A and B changed according to the identity of sample stimulus.

Both monkeys performed the task with high accuracy (*Figure 2A*) as more than 85% of trials were hits, less than 10% were misses, and less than 5% were false-alarms. Moreover, analyses of stimuli triggering the false-alarm responses (*Figure 2B*) revealed that monkeys correctly ignored test stimulus A during sample B trials and test stimulus B during sample A trials.

## Electrophysiology

We recorded the activity of 201 individual LIP neurons while monkeys performed the tasks (74 during two-location and 127 during one-location DCM tasks). We analyzed correct trials only. Our task design allowed us to test how sensory content (i.e. identity of the test stimuli) and behavioral relevance (i.e. match status of test stimuli) jointly shaped LIP neuronal selectivity. We will also briefly describe the influence of sample identity on the LIP neuronal response. In addition, results will be presented separately for each monkey in each figure but grouped together for statistical tests. However, unless mentioned otherwise, analyzing data separately for each monkey did not affect the outcomes of the following analyses.

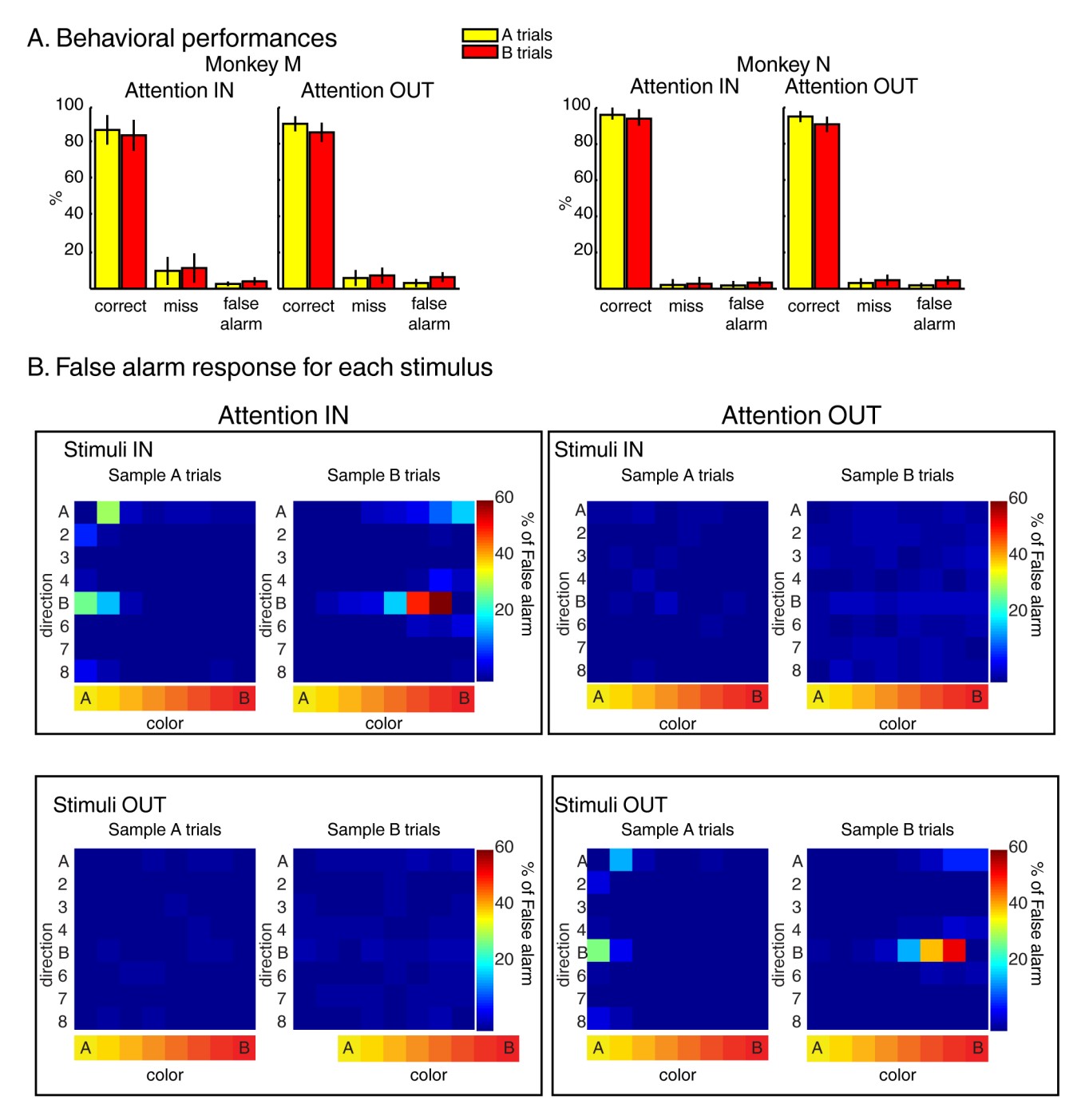

**Figure 2.** Behavior during the two-location DCM. (**A**) Performances: both monkeys M and N performed the task with high accuracy as ~90% of trials were correct, ~6% were misses and ~4% were false alarms (excluding fixation breaks) in both attention IN and attention OUT conditions. (**B**) Percent of false alarm responses (averaged across both monkeys) for each of the 64 test stimuli located inside (top) or outside (bottom) the RF of the recorded neuron during attention IN (left) and attention OUT (right) conditions. For each panel, each row represents one direction, and each column represents one color.

## Typical LIP neuronal response

*Figure 3* shows the activity of four different neurons with different patterns of selectivity for the identity, match status, and manual response evoked by test stimuli. Selectivity to the identity of stimuli is exemplified by neuron #1 (*Figure 3A*, left panel). This neuron responded preferentially to test stimulus B independently (at least in the initial phase of the response) of its match status. Interestingly, the relative strength of identity and match selectivity evolved in time. This neuron was identity-selective from 86 ms to 299 ms after stimulus onset (*Figure 3*, black horizontal line in the ROC panel, Wilcoxon test, Bonferroni corrected, p<0.01) and became selective to the match status of the stimuli 209 ms after stimulus onset (*Figure 3*, grey horizontal line in the ROC panel, Wilcoxon test, Bonferroni corrected, p<0.01), shortly before this monkey's average RT. Example neuron #2 (*Figure 3B*) illustrates selectivity to the match status of the stimuli (starting 53 ms after stimulus onset), independent of their identity (left panel). The absence of selectivity to target stimuli located

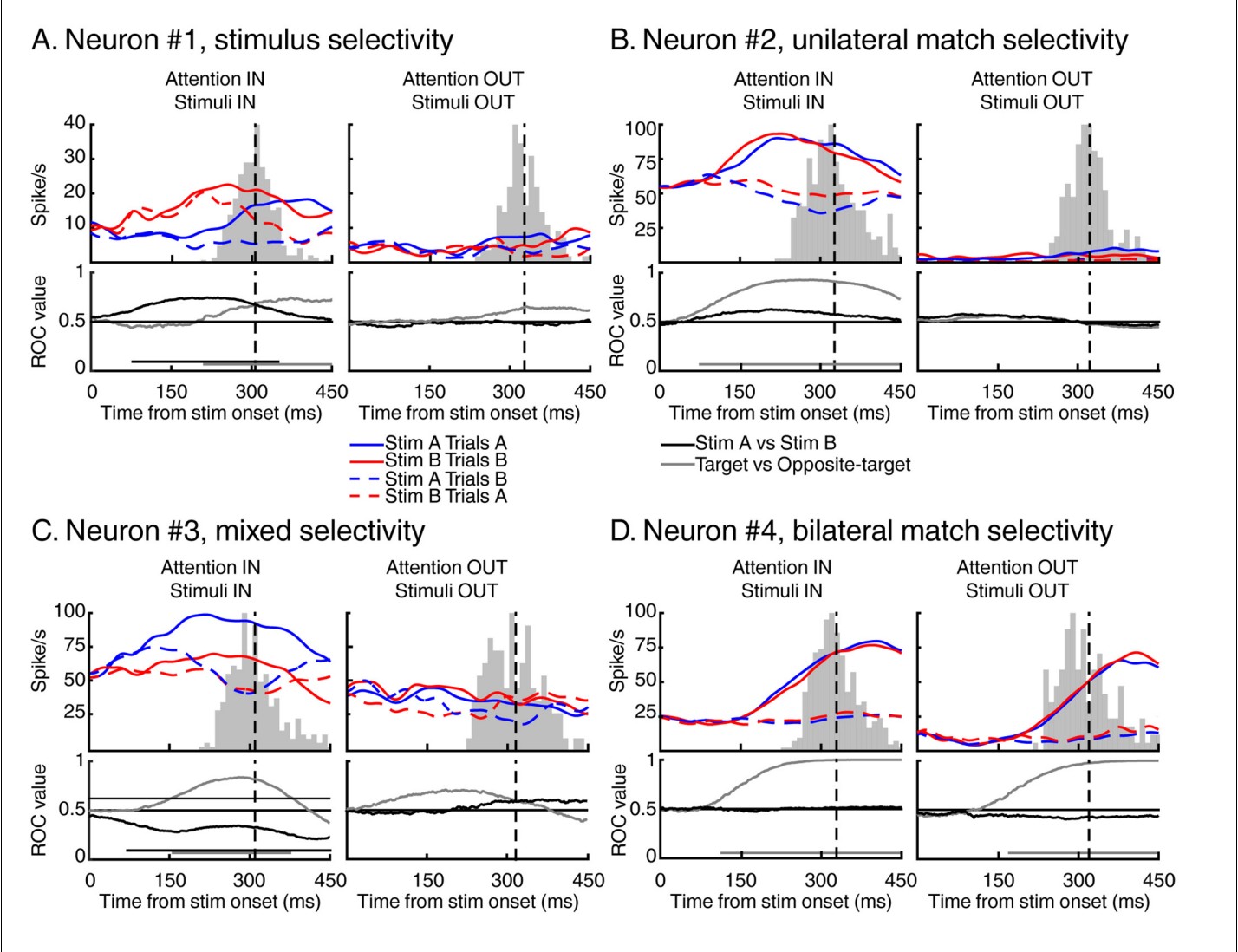

**Figure 3.** Examples of individual LIP neurons. For each example, we show the time course of response to test stimulus A (in blue) and test stimulus B (in red) for both target stimuli (full lines) and opposite-target stimuli (dashed lines). Vertical black dashed lines represent averaged monkey's reaction times, surrounding grey histograms represent RT's distributions. We also show the time course of identity selectivity (black lines, ROC comparison stimulus A vs stimulus B) and of match selectivity (grey lines, ROC comparison target vs opposite-target). Each example represents a stereotypical response observed among our recorded population of 74 LIP neurons.

outside this neuron's RF (right panel) suggests that such selectivity cannot be attributed to a motor-preparation signal. However, this pattern of selectivity could also reflect motor-preparation signals modulated by cognitive factors such as the position of attention.

Moreover, selectivity for the identity and match status of test stimuli were not mutually exclusive and some neurons multiplexed these signals. Neuron #3 (*Figure 3C*) responded preferentially to test stimulus A compared to test stimulus B (starting from 86 ms after stimulus onset) but also to target stimuli compared to opposite-target stimuli (starting from 124 ms after stimulus onset). We also observed neurons which showed bilateral selectivity to match stimuli, meaning that they responded more strongly to target stimuli (which required a manual response) in either the ipsilateral or contra-lateral visual field. For example, neuron #4 (*Figure 3D*) responded preferentially to target compared to opposite-target stimuli (starting 103 ms and 171 ms after stimulus onset during attention IN and attention OUT respectively), consistent with several processes such as spatially independent match/non-match decisions, preparation of the manual movement used to report the presence of match stimuli, or reward expectation. Although we treated bilateral and unilateral match selectivity similarly in our analyses, we conducted a series of control analyses (*Figure 4—figure supplement 2*, Figure 6, *Figure 7—figure supplement 1*) which show that removing bilateral selectivity from the two-location DCM dataset yielded qualitatively similar findings.

In the following, we will characterize these two types of selectivity (identity and match status) at the population level along with the influence of sample identity on LIP neuronal responses.

## Stimulus identity and match selectivity

We characterized stimulus identity and match selectivity using a receiver operating characteristic (ROC) analysis. We compared each neuron's responses either to test stimuli A and B (identity selectivity), or to target stimuli and opposite-target stimuli (match selectivity)—in both cases for stimuli located inside neurons' RF. But, as shown in previous studies (*Ibos and Freedman, 2014*, *2016*), LIP neuronal activity was strongly influenced by the identity of the sample stimulus (Figure 7C) during sample presentation but also during the delay period and prior/during the presentation of each test stimulus. Sample encoding during the delay and test periods is consistent with a role for LIP in encoding task-relevant information in short-term memory. This characteristic of the neuronal response in LIP interfered with our analysis of the dynamics of sensory selectivity as it was responsible for larger response during either sample A or sample B trials, mimicking the effect of identity selectivity on the ROC analysis prior stimulus onset in several neurons (N = 24/74, Wilcoxon test, p<0.05). Therefore, prior to analyzing data, we equated each neuron's response during either sample A or B trials (depending on each neuron's preference) using a randomized decimation approach (an approach more conservative of spike train dynamics than standard normalization methods) based on the ratio of firing rates between sample A and sample B trials prior to test stimulus onset (see Materials and methods). After this correction, we observed that a large fraction of LIP neurons (N = 147/201; sliding Wilcoxon test (200 ms window), Bonferroni corrected, p<0.01) showed selectivity for the identity (N = 92; 51/67 for monkey N, 41/70 for monkey M, *Figure 4—figure supplement 1*) and/or the match status of the stimuli (N = 103; 34/67 for monkey N, 69/80 for monkey M). In addition, 48/147 neurons (18/67 for monkey N, 30/80 for monkey M) showed significant selectivity for both types of information. This shows that both types of information (identity and match selectivity) are encoded by LIP neurons from both monkeys. *Figure 4A* shows each neuron's (N = 147) time-course of both identity (left panel) and match selectivity (middle panel). Neurons are sorted according to the averaged value (100 to 350 ms after stimulus onset) of a visuo-decision index comparing outcomes of the aforementioned ROC analyses (VDI, identity selectivity – match selectivity, *Figure 4A*, right panel). The VDI indicates that the population of LIP neurons selective to either the identity of test stimuli and/or their match-status (N = 147) forms a continuum from purely identity-selective neurons (VDI = 1) to mixed identity and match/non-match selectivity (VDI~ = 0) to purely match-selective neurons (VDI=−1). Neurons from both monkey M and N showed identity and match selectivity. However, the respective strength of selectivity differed between the monkeys (*Figure 4—figure supplement 1*): distribution of VDI from monkey M are shifted toward negative values (match selectivity) and toward positive values for monkey N (identity selectivity).

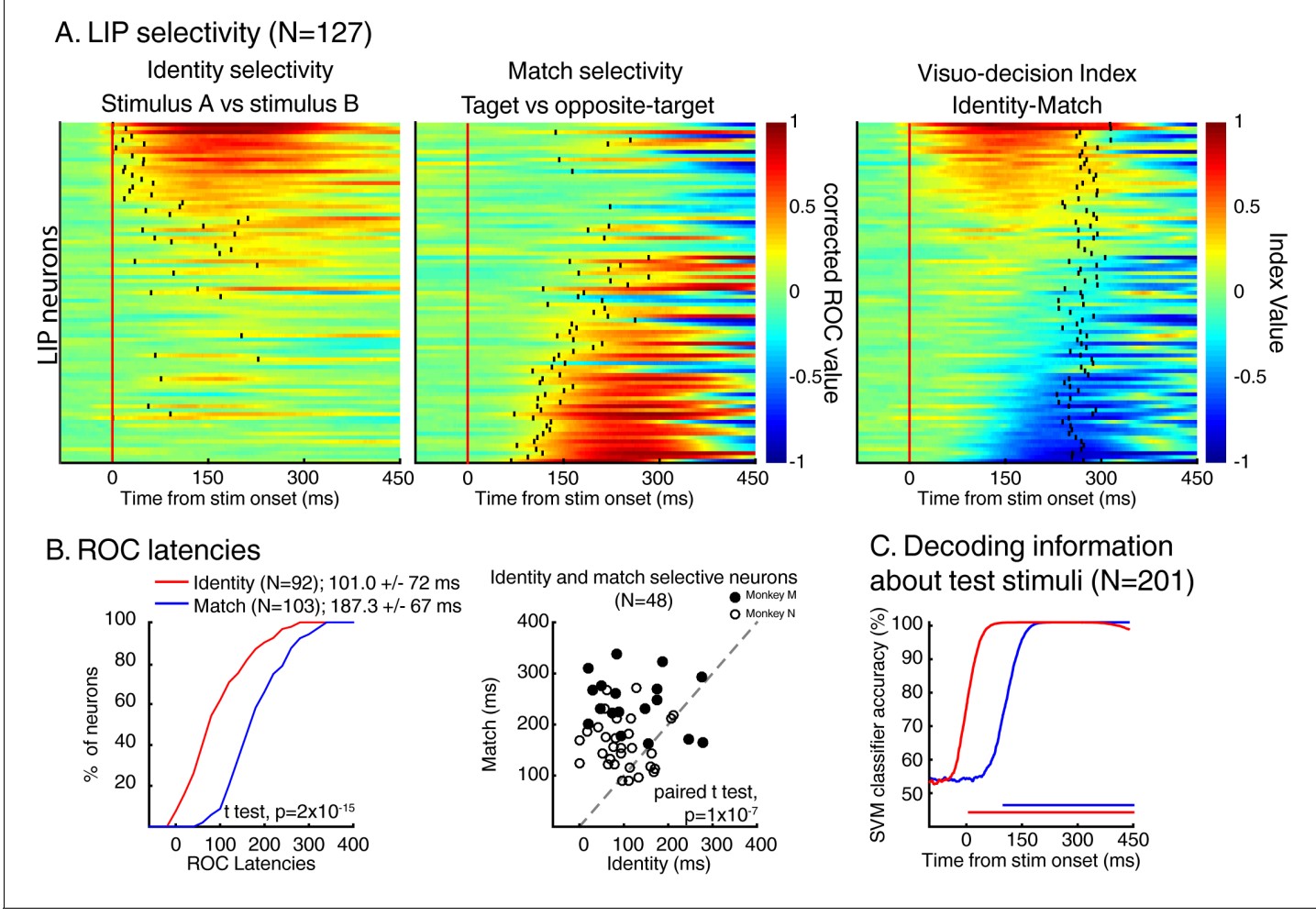

**Figure 4.** Dynamics of identity and match selectivity in LIP. (A) Each line of each of these panels correspond to the time course of identity selectivity (left panel) and match selectivity (middle panel), defined as corrected ROC values. Each vertical black tick represents the latency at which each signal becomes statistically significant (Wilcoxon test, corrected for multiple comparisons, p<0.01). Right panel shows the time course of a visuo-decision index for each LIP neuron. Vertical black ticks represent averaged RT. Lines on each panel represent similar neurons. (B) Comparisons of discrimination latencies for identity and match selectivity from both two-location and one-location datasets. (C) Time course of performances of an SVM classifier to decode the identity (red line) and the match status of test stimuli (blue line) of both DCM tasks (N = 201). Horizontal blue and red lines represent the statistical significance of the decoder performance (p<0.01).

The following figure supplements are available for figure 4:

**Figure supplement 1.** The same analysis as in *Figure 4A and B* but for each dataset and for monkey N and M separately.

**Figure supplement 2.** The same analysis as in *Figure 4* but for the two locations dataset only after removing bilaterally match selective neurons.

## Dynamics of neuronal selectivity

As shown by the example neurons, LIP expressed patterns of identity and match/non-match selectivity with different dynamics. Three complementary methods revealed that LIP neurons encoded the identity of test stimuli before their match/non-match status. First, a direct comparison of each signal's latency (black vertical ticks in *Figure 4A* left and middle panel, see Materials and methods) revealed that identity-selectivity (N = 92, mean latency = 101.0 ms ± 72 ms) rose significantly earlier than match-selectivity (N = 103, mean latency = 187.3 ms ± 67 ms; t test, p=2*10$^{-15}$; *Figure 4B*, left panel). This is not due to higher firing rates of identity-selective neurons (N = 92) compared to match-selective neurons (N = 103) since paired comparisons of latencies for neurons selective for

both signals (N = 48) revealed a similar dynamic (mean latency: stimulus identity: 107.8 ms ± 68 ms; match selectivity: 191.6 ms ± 65 ms; paired t test, p=1×10⁻⁷). We confirmed this observation by using a population decoding approach in which we trained SVM classifiers to decode independently the identity or the match status of the stimuli based on the activity of the entire population of LIP neurons (N = 201, see Materials and methods). We found that linear classifiers could decode, above the level expected by chance (50%), the identity of test stimuli (0 ms after stimulus onset) prior to their match status (93 ms after stimulus onset; *Figure 4C*). This serial representation of identity and target status in LIP population suggests a role in transforming sensory information into a decision about the match status of test stimuli.

To test this hypothesis, we analyzed whether the time course of each neuron's activity was correlated to the monkeys' RTs (*Figure 5*). For each neuron, we paired the time course of each trial's z-scored firing rate (convolved with a Gaussian function, σ = 15 ms) with that trial's RT. Importantly, for this analysis, we only used trials in which the sample stimulus was each neuron's preferred

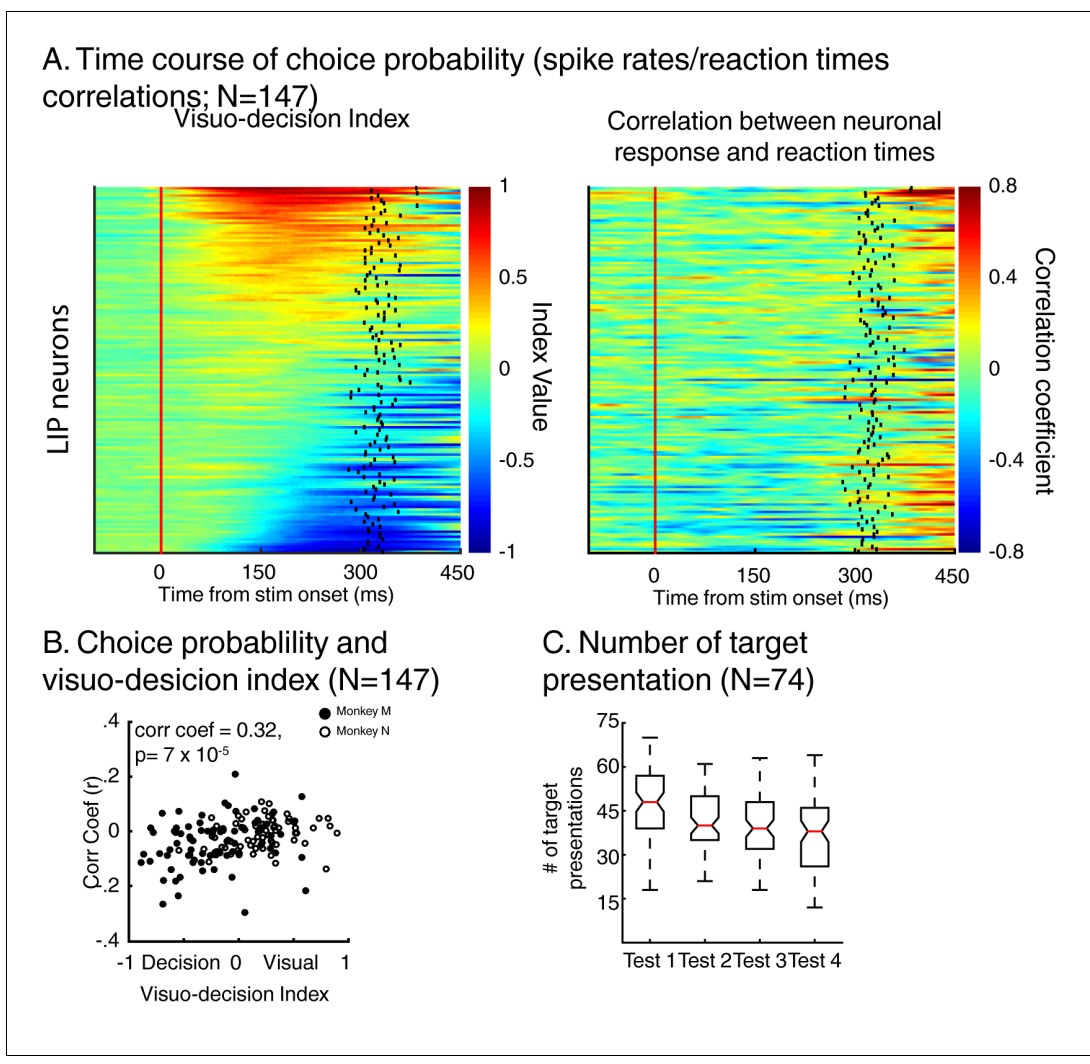

**Figure 5.** Dynamics of neuronal response and monkeys' reaction times. (A) Left: time course of visuo-decision index (similar as *Figure 4A*, N = 147). Right: corresponding time course of choice probability (correlation between neuronal responses (z-scores) and behavioral reaction times). (B) Correlation between choice probability and visuo-decision index. (C) Number of target presentation (two-location DCM only) as a function of target's position in the sequence of test stimuli. Red lines represent the median of the distribution, superior and inferior limits of the boxes represent the edges of the 25th and 75th percentiles, error bars represent the most extreme values of the distributions.

stimulus (e.g. sample A trials for neurons preferring test stimulus A). Artefactual correlation in this kind of analyses can emerge when monkeys' detection rates fluctuate with the timing at which the stimuli were presented during each trial (*Kang and Maunsell, 2012*), leading to more presentations of the target stimulus at a certain trial epoch (e.g. a larger number of targets detected during the presentation of the first test stimulus). In this situation, an important control is needed to ensure that the results of this analysis (correlation between firing rates and behavioral reaction times) do not follow a similar pattern. To do so, we first measured whether the number of presentation of the target stimulus depended on its position in the sequence of test stimuli. We observed an over-presentation of target during the first test period compared to the three successive ones (*Figure 5C*, ANOVA, $p=2\times10^{-7}$). Therefore, we removed data acquired during presentation of target stimuli during the first test period from our analysis, using only test stimulus presentations 2, 3 and 4 (2 and 3 for the one-location DCM task). At the population level (N = 147), the magnitude of this correlation (r) covaried with each neuron's VDI (*Figure 5B*, correlation coefficient = 0.32, $p=7\times10^{-5}$). The stronger the influence of match-selectivity compared to identity selectivity, the larger the negative correlation, suggesting that match-selectivity, but not identity-selectivity, reflects a process which covaries with monkeys' decisions about the relevance of the stimuli.

This correlation between the dynamics of match selectivity with monkeys' reaction times could, for example, reflect decision-related (as shown in *Figure 3B*) or non-spatial processing such as motor-preparation signals (as exemplified in *Figure 3D*). To address the potential confound between unilateral and bilateral match selectivity, we compared the degree of selectivity to target stimuli located inside and outside neurons' RFs in the two-location protocol only (*Figure 6*). However, as shown in a previous study (*Ibos and Freedman, 2016*), SBA strongly influenced the activity of LIP neurons, with larger activity when monkeys attended inside LIP neurons' RFs. We sought to remove the influence of SBA on match-selectivity by equating the level of average neuronal activity between attention IN and attention OUT condition. To do so, we used a decimation approach based on the firing rate ratio (Attention IN/Attention OUT) during pre-test stimuli activity (200 ms prior to test stimulus onset, see Materials and methods). A laterality index (target/opposite-target IN minus target/opposite-target OUT), showed that non-spatial processes likely influenced the response of a sub-population of LIP neurons. Laterality index values close to 0 reveal selectivity for target stimuli located inside and outside each neuron's RF, while positive or negative values indicate match-selectivity limited to stimuli located inside or outside RFs. A permutation test (see Materials and methods) revealed that bilateral match selectivity signals could account for the match selectivity of 20/41 neurons (*Figure 6B*, permutation test, p<0.05) as they were selective independently of the spatial location of match stimuli. Removing these 20 bilaterally match selective neurons from the pool of 60 neurons in *Figure 5A* did not change the finding of an inverse correlation between choice probability (r-values, linking each neuron's dynamics and behavioral reaction times) and VDI (*Figure 6C*, p=0.01). Note that when separating monkey M and N, this effect is significant for monkey M (N = 15, p=0.049) but not for monkey N (N = 25, p=0.9). However, our task was not specifically designed to characterize the exact nature of non-spatial match-selectivity signals, which could reflect different processes (e.g. non-spatial decisions, motor-preparation, motor-execution or even reward expectancy). Nevertheless, we were able to identify a population of LIP neurons showing match-selectivity that was independent of non-spatial processes (including motor-preparation), suggesting a role in decision-related computation independent of motor-related processing.

## Encoding of visual feature conjunctions

In the previous section, we showed that a population of LIP neurons encode the identity of specific conjunctions of visual features (color and motion-direction). We also showed in previous reports (*Ibos and Freedman, 2014*, *2016*) that the same LIP neurons independently encode the color and motion-direction of these stimuli in a manner consistent with a bottom-up integration model. But this model does not account for how LIP neurons combine signals related to color and motion-direction to encode either the identity or the match status of test stimuli. We focus on how LIP neurons encode target stimuli (e.g. test A for stimulus A selective neurons) compared to color-match stimuli (conjunction of the relevant color with any of the seven irrelevant directions), to direction-match stimuli (conjunction of the relevant direction with any of the seven irrelevant colors) and to opposite-target stimuli (e.g. test-stimulus A during sample B trials for A selective cells). The averaged normalized firing rate (N = 147/201, *Figure 7A*, top left panel) to each of these stimuli reveals a gradient of

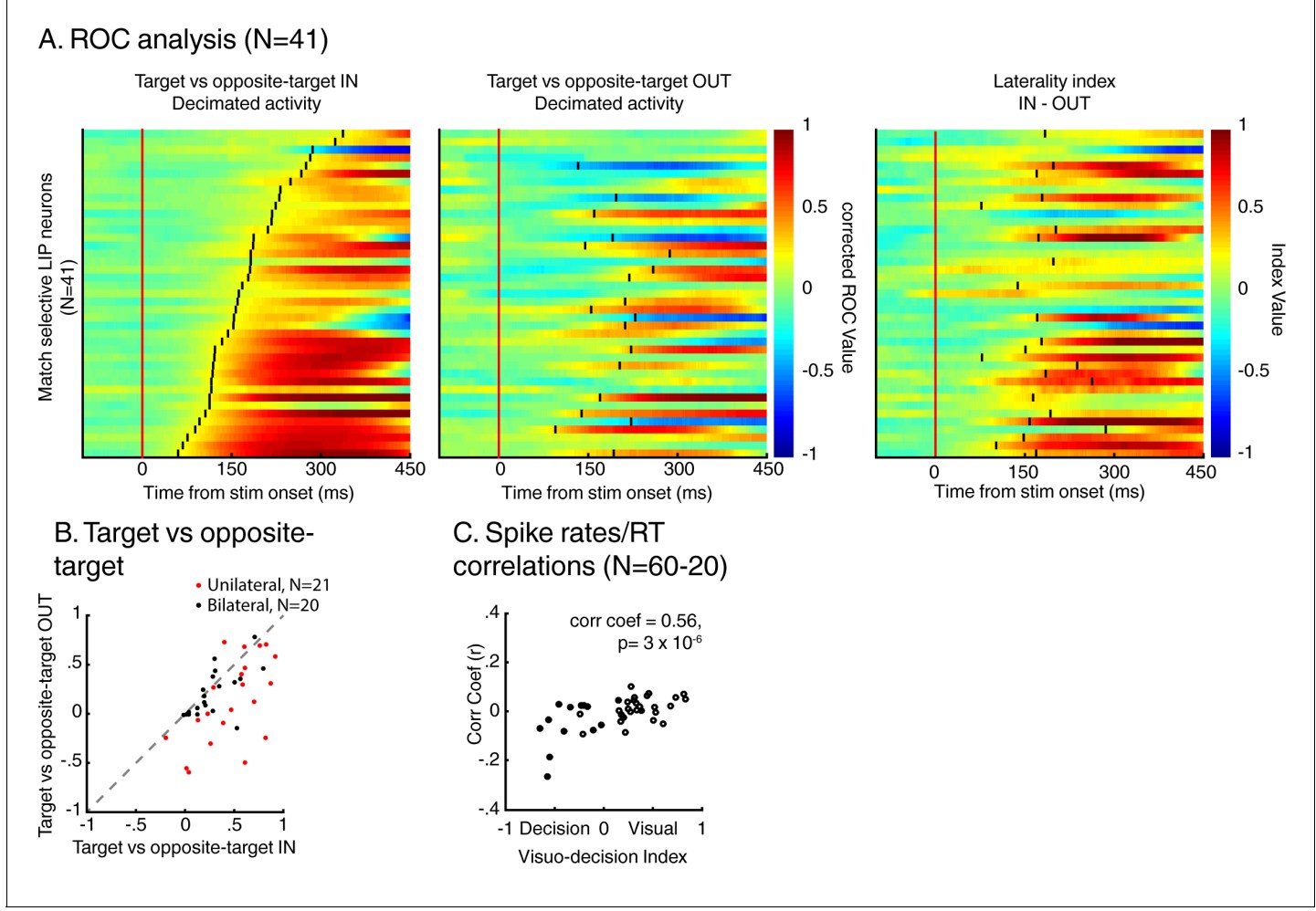

**Figure 6.** Spatial selectivity of decision-related signals (two-location DCM only). (**A**) Each line in both the left and middle panels correspond to the time course, for one LIP neuron, of match selectivity when relevant stimuli were located inside (left panel) or outside its RF (middle panel). Vertical black ticks correspond to each signal latency of significance (Wilcoxon test, corrected for multiple comparisons, p<0.01). Right panel depicts the time course of a laterality index. Vertical black ticks correspond to the latency at which IN and OUT signals significantly differ (permutation test corrected for multiple comparison, p<0.01). (**B**) Comparisons of match selectivity during attention IN and attention OUT conditions. (**C**) Similar to B but with baseline activity equated between attention IN and attention OUT. (**C**) Correlation of neuronal dynamic and monkeys' reaction times similar as *Figure 5B* after removing neurons bilaterally match selective.

responses: target and opposite-target stimuli triggered the largest and smallest responses, respectively, while color and direction-match stimuli triggered intermediate responses. In the framework of our task, responses to non-match stimuli represent the encoding of irrelevant information (e.g. irrelevant features). Therefore, in order to test how LIP neurons combine relevant information, we removed signals related to irrelevant information by subtracting, for each neuron, the response to opposite-target stimuli from the responses to target, color-match and direction-match stimuli (see *methods* for details). Then, we calculated each neuron's *additivity-index* (*Additivity-index*=target − (color + direction)). Despite its name, this index does not reflect purely linear processes since it results from subtractive normalization (a non-linear computation) in order to take into account relevant information only. This method assumes that irrelevant information contained in target, color-match and direction-match stimuli are equivalent to opposite-target stimuli. An *additivity-index* of 0 corresponds to a pseudo-linear model in which neurons encode conjunctions of features by summing feature-specific sensory inputs. Positive and negative *additivity-indices* represent super-additive and sub-additive mechanisms, respectively. This *additivity-index* (*Figure 7A*, right panel) was significantly

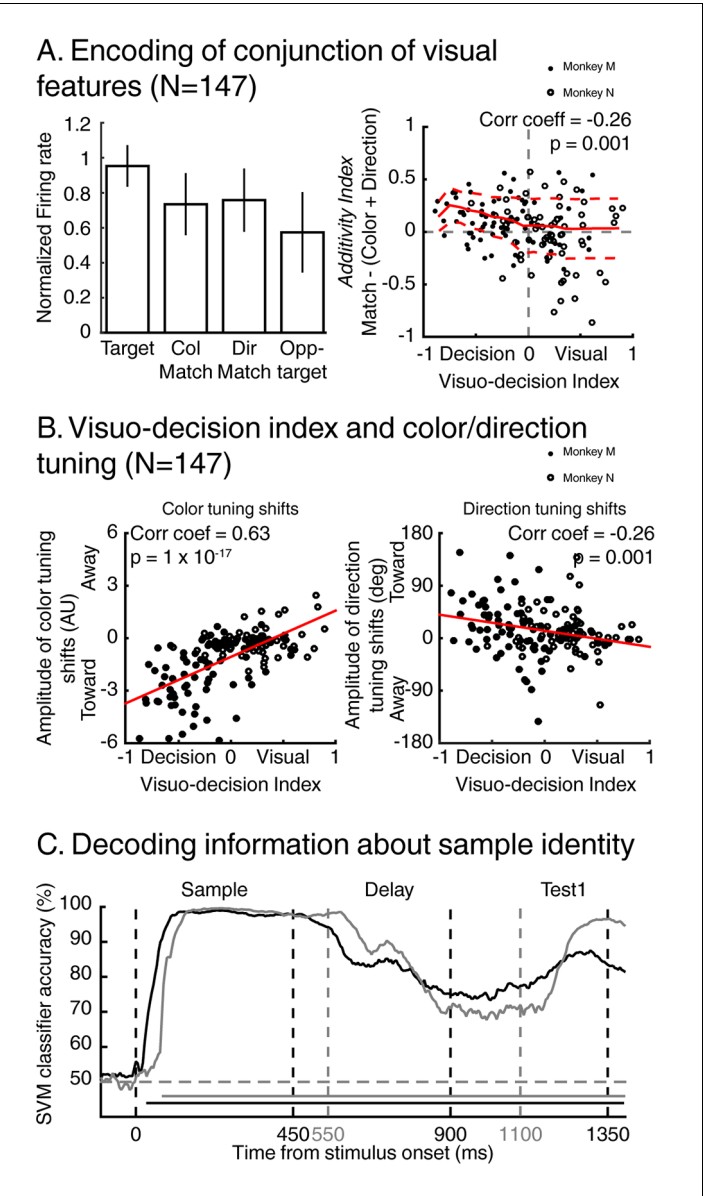

**Figure 7.** Encoding of visual features along the visuo-decision continuum. (A) Left panel: averaged normalized response of neurons from the visuo-decision continuum (N = 147) to target, color-match, direction-match and opposite target stimuli. Error bars represent standard deviation of the mean (std). Right panel: Response to relevant conjunction (match) are compared to linear model in which neurons combine color and direction signals linearly (color+direction) as a function of averaged visuo-decision index. Full red line represents sliding averaged additivity index, dotted red lines represent standard deviation to the mean. (B) Effect of feature-based attention as a function of VDI on color selectivity (left, negative and positive values represent shifts toward and away the relevant color respectively), and on direction selectivity (right, positive and negative values represent shifts toward and away the relevant direction, respectively). (C) Time course of decoding accuracy of the identity of the sample by a linear SVM classifier using the activity of the entire population of either 74 LIP neurons (black lines, two-location DCM; chance level = 50%) or 127 LIP neurons (grey lines, one-location DCM; chance level = 50%).

The following figure supplement is available for figure 7:

**Figure supplement 1.** The same analysis as in *Figure 7* but applied to the two -location dataset after removing bilaterally match selective neurons.

negatively correlated with each neuron's VDI (correlation coefficient = $-0.26$, p=0.001). This relationship was impacted by separating data from each monkey only when we also removed bilaterally match selective neurons (monkey M, N = 15: correlation coefficient = $-0.37$, p=0.18; monkey N, N = 25, correlation coefficient = $-0.22$, p=0.28), likely due to the decrease in sample size. Visual (positive VDI) neurons' additivity-indices were centered on 0, reflecting additive encoding of conjunction of features, suggesting that identity-selective neurons express their selectivity by linearly pooling bottom-up color and direction encoding. On the contrary, decision-related neurons (negative VDI) showed apparent super-additive (positive additivity-index) encoding of conjunction of features. This was expected given that the information encoded by these neurons is not exclusively related to sensory information, but it is not clear whether and how match-selective neurons integrate and pool bottom-up sensory information in order to generate their selectivity.

In order to understand the influence of bottom-up sensory signals on LIP match-selectivity, we tested the impact of FBA on each neuron along the visuo-decision continuum. We previously showed that FBA shifts LIP's representation of individual features toward the relevant direction and color (*Ibos and Freedman, 2014*, *2016*) in a manner consistent with bottom-up integration of sensory information. Moreover, we showed that the amplitude of these feature-tuning shifts depended on the strength of each LIP neuron's feature-selectivity, which presumably reflects the connectivity between individual LIP neurons and upstream cortical visual neurons. We posited that visually selective LIP neurons (tuned to visual features) are less likely to be modulated by attention than decision-related neurons. To examine this, we compared each neuron's amplitude of feature-tuning shifts (see Materials and methods) with its respective VDI (*Figure 7B*). As predicted, we found a significant correlation between each neuron's VDI and the amplitude of feature-tuning shifts (for color: correlation coefficient = 0.63, p=$1\times10^{-17}$; for direction: correlation coefficient = $-0.26$, p=0.001), showing that FBA had a larger impact on neurons involved in encoding decision-related signals than on visually selective neurons. Difference in the sign of these correlations results from the respective methods used to describe color and direction tuning shifts (see Materials and methods). For this analysis, separating data from each monkey affected the results when considering the relationship between the effects of FBA on direction selectivity and VDI. This was true for monkey N only when considering both datasets (correlation coefficient = 0.08, p=0.53) and for both monkeys when removing neurons showing bilateral match selectivity (p>0.05 for both monkeys). This was not surprising given the subtle effects of FBA on LIP direction selectivity (*Ibos and Freedman, 2014*, *2016*) described previously. This relationship suggests that match-selectivity in LIP is a composite signal resulting from the pooling of bottom-up sensory information with additional extraretinal signals. Whether it reflects local computation or integration of top-down signal originating in downstream cortical areas is an important question to be considered in future work.

## Discussion

We recorded the activity of LIP neurons from two monkeys performing a DCM task. Monkeys released a touch bar when test stimuli matched the position, color and motion-direction of a previously presented sample stimulus. On each trial, sample stimulus was randomly picked among two different stimuli and two different positions. Therefore, depending on the identity of the sample, these conjunctions could either be target or opposite-target stimuli. This task design allowed us to characterize the relative importance of sensory and cognitive information in the response of LIP neurons to test stimuli. This revealed that LIP neurons multiplexed both the identity and the match status of stimuli. Interestingly, sensory information (whose time course was weakly correlated with monkeys' RTs) was encoded prior to decision-related signals (whose timing was predictive of monkeys' RTs). Finally, we studied how LIP neurons combine color and motion-direction signals in order to encode their conjunction. We found that, along the visuo-decision continuum, the encoding of the relevant conjunctions switched from additive to super-additive. It suggests that (1) identity-selective neurons primarily integrate only bottom-up sensory information, and (2) match-selective neurons integrate additional sources of information. Several previous studies described aspects of LIP activity which could relate to the selectivity described here. For example, LIP identity selectivity relates to selectivity to the category membership of test stimuli described in some previous work (*Freedman and Assad, 2006*). Similarly, selectivity to behaviorally relevant stimuli has been shown in LIP by different studies (*Bisley and Goldberg, 2003*; *Leathers and Olson, 2012*;

*Swaminathan et al., 2013*; *Ibos et al., 2013*). However, the current study is the first to directly characterize and compare each of these types of selectivity in LIP and to integrate them in a coherent theoretical framework. We propose that LIP is composed of a visuo-decision continuum of neurons involved to different degrees in (1) the integration of bottom-up sensory inputs, (2) the linear grouping of these disparate sensory signals and (3) their transformation into a signal related to monkeys' decisions about the relevance of the visual stimuli. This transformation could result from either local computation, or from the integration of additional top-down signals, but the nature of such computations is still unknown (see below).

## Previous model of decision-making in LIP

In the past decades, posterior parietal cortex has been associated with certain aspects of decision-making by a large corpus of studies (*Leon and Shadlen, 1998*; *Shadlen and Newsome, 2001*; *Gold and Shadlen, 2007*; *Huk and Shadlen, 2005*). These studies employed similar paradigms in which monkeys discriminated the direction of movement of noisy stimuli and reported their decisions with a saccadic eye movement. During presentation of the noisy-stimuli, if the target of the upcoming saccade was positioned in neurons' RFs, average responses in LIP monotonically increased (but see [*Latimer et al., 2015*]) until they reached a threshold, which corresponded to the execution of the eye-movement. It was posited that the ramping activity reflects accumulation of sensory evidence leading to the decision about the direction of the saccade. This interpretation is at least semantically similar to our observations that LIP, at the population level, appears to integrate, combine and transform sensory information into decision-related signals.

However, despite these similarities, our approach and model differ in fundamental ways in terms of mechanisms underlying the transformation of decisions into task-appropriate actions. As noted by several studies (*Filimon et al., 2013*; *Freedman and Assad, 2016*), the stereotypical ramping activity preceding the execution of saccadic eye movement (*Leon and Shadlen, 1998*; *Shadlen and Newsome, 2001*; *Gold and Shadlen, 2007*; *Huk and Shadlen, 2005*) is likely to be driven to a large degree by processes related to preparing the upcoming saccade—which have been extensively shown to modulate the activity of LIP neurons (*Gnadt and Andersen, 1988*; *Barash et al., 1991*)—rather than by the motion stimulus or by the accumulation of sensory evidence. This is especially evident since the pre-saccadic activity of LIP neurons has long been associated with the intention of performing a saccadic eye movement toward the RF (*Gnadt and Andersen, 1988*; *Andersen and Buneo, 2002*; *Bracewell et al., 1996*; *Mazzoni et al., 1996*), that reversible inactivation of LIP does not alter monkeys' ability to perform such sensorimotor transformation (*Katz et al., 2016*), and since the motion stimulus in this task is presented outside LIP neurons' RFs (*Freedman and Assad, 2016*).

Another key difference between our work and previous studies on perceptual-decisions is related to the definition of decision-making. Previous work proposed that perceptual-decisions regarding a visual stimulus are supported by the same neurons as the ones encoding motor actions resulting from its detection (*Gold and Shadlen, 2007*; *Cisek and Kalaska, 2010*). However, this claim is a subject of debate (*Filimon et al., 2013*; *Freedman and Assad, 2016*; *Tosoni et al., 2008*). Our study, along with others (*Leathers and Olson, 2012*; *Filimon et al., 2013*; *Freedman and Assad, 2016*), distinguishes between perceptual-decision-making (making a decision about a sensory event) and motor-decision-making (making a decision about which action to initiate). For example, deciding whether an animal is a predator is different from deciding whether the best survival strategy is to run away or to attack him in order to defend your offspring. Our approach is supported by studies showing that LIP multiplexes and independently encodes cognitive (abstract category-selectivity) and saccade-related signals (*Rishel et al., 2013*). Although our task was not designed to specifically distinguish between detection, decision-making and motor-preparation, a large fraction of LIP neurons encoded the relevance of stimuli independently of monkeys' behavioral responses. This supports the hypothesis that LIP is involved in perceptual decision-making independent of action.

## Toward a new model of decision-making in LIP?

Recently, we showed that representation of individual features in LIP is consistent with the linear bottom-up integration of the activity of populations of feature-selective neurons in upstream visual areas (*Ibos and Freedman, 2014*, *2016*). However, this model does not account for how and whether LIP neurons flexibly combine these representations together in order to encode the identity and match-

status of test stimuli. Testing how LIP neurons combine signals related to each feature allowed us to begin to identify which computations take place along the LIP visuo-decision continuum. We propose here that identity-selective neurons linearly pool color and motion-direction selective signals, presumably originating from upstream cortical areas. According to Bayesian inference theory (*Kersten et al., 2004*), decision-variables (match-selectivity in our data) are computed by comparing statistical inferences (*Parker and Newsome, 1998*) (identity-selectivity in our data) about the stimuli subjects are looking at to an internal model of the stimuli subjects are looking for. The super-additivity observed for match-selective neurons in LIP hypothetically is the outcome of a computation related to the comparison of the identity of the stimuli with top-down signals, presumably originating in PFC (*Mante et al., 2013*), related to internal models of which features are behaviorally relevant. The selectivity for sample identity across each trial epoch (*Figure 7C*) could reflect the integration of top-down signals related to the identity of the relevant stimuli kept in working memory. Unfortunately, we still lack evidence to understand whether this match-selectivity results from local computation or from the integration of signals from another source. Therefore, future research should focus on developing and testing this question. Specifically, it will be interesting to test how LIP neurons along the visuo-decision continuum respond to independent information encoded in different brain areas—including noisy sensory information, diverse expected value and behavioral costs—and how these sensory and cognitive representations are combined and transformed into a decision signal in LIP.

Finally, we showed that LIP encodes all the information required to solve the DCM task: (1) the information being encoded in working memory; (2) the stimuli monkeys were looking for; (3) the stimuli monkeys were currently looking at; (4) the monkeys' match/non-match decisions; (5) the monkeys' behavioral responses. This large amount of mixed selectivity poses a decoding challenge for downstream areas in charge of reading-out the activity of LIP neurons. Our results underline the diversity of signals influencing and shaping LIP neuronal selectivity and urge us to characterize the functional connectivity that allow LIP neurons to mediate decision-making by facilitating interactions between both upstream and downstream areas.

## Materials and methods

### Behavioral task and stimulus display

Experimental procedures were identical to the ones presented in a previous report (*Ibos and Freedman, 2014*, *2016*). The amplitude of feature-tuning shifts were already presented in this report. Two male monkeys (*Macaca mulatta*, monkey M, ~10 kg; monkey N, ~11 kg) were facing a 21 inch CRT monitor on which stimuli were presented (1280*1024 resolution, refresh rate 85 Hz, 57 cm viewing distance), seating head restrained in a primate chair inserted inside an isolation box (Crist Instrument). Stimuli were 6° diameter circular patches of 476 random colored dots moving at a speed of 10°/s with 100% coherence. All stimuli were generated using the LAB color space (1976 CIE L*a*b), and all colors were measured as isoluminant in experimental condition using a luminance meter (Minolta).

Gaze position was measured with an optical eye tracker (SR Research) at 1.0 kHz sample rate. Reward delivery, stimulus presentation, behavioral signals and task events were controlled by MonkeyLogic software (*Asaad et al., 2013*), running under MATLAB on a Windows-based PC.

### Electrophysiological procedures

Monkeys were implanted with a headpost and recording chamber during aseptic procedures. Both the stereotaxic coordinates and the angle of the chambers were determined by 3D anatomical images obtained by magnetic resonance imaging conducted prior to surgery. The recording chambers were positioned over the left intraparietal sulcus. All procedures were in accordance with the University of Chicago's Animal Care and Use Committee and US National Institutes of Health guidelines.

During each experiment session, a single 75 µm diameter tungsten microelectrode (FHC) was lowered into the cortex using a motorized microdrive (NAN Instruments) and dura-piercing guide tube. Neurophysiological signals were amplified, digitized and stored for offline spike sorting (Plexon) and analysis.

### Data analysis

We analyzed neuronal activity acquired during correct trials. Behavioral and neurophysiological results were similar in both monkeys, allowing us to merge datasets for population analysis.

### Decoding sample's identity

We trained and tested a linear support vector machine classifier (*Chang and Lin, 2011*) to decode the identity of the sample stimulus based on the neuronal response during sample presentation, delay period and presentation of the first test stimulus located inside neurons' RF. Training sets were built by randomly picking with replacement 70 trials from the pool of sample A trials, and 70 trials from the pool of sample B trials. Testing sets were built by picking with replacement 30 trials, different from training trials, for similar conditions each. This procedure was repeated 1000 times, and classifier was considered to perform above chance (50%) if the accuracy of the decoder was higher than chance level for more than 990/1000 iterations (p<0.01).

### Identity and match selectivity

Two sliding Receiver Operating Characteristic (ROC) analyses (200 ms analysis window sliding in millisecond steps around test stimulus onset) were used to characterize LIP neuronal selectivity to the identity of test stimuli A and B and to their match status. Identity selectivity was characterized by comparing neuronal firing rates to test stimulus A to test stimulus B (whether stimuli were target or opposite-target stimuli). Match selectivity was characterized by comparing neuronal firing rates to target and opposite-target stimuli. Significance of each comparison was defined using non-parametric Wilcoxon-test corrected for multiple comparisons (p<0.01, Bonferroni corrected).

### Dynamics of neuronal responses

*Individual neuron latencies:* We defined neuronal latencies for identity-selectivity and match-selectivity independently as the first time at which the above-mentioned sliding Wilcoxon-tests reached significance (p<0.01, Bonferroni corrected) for 100 consecutive milliseconds. We limited our analysis to a window between the onset of the test stimuli and the averaged manual-RT during the respective recording session. *Decoding:* We trained and tested two independent linear support vector machine classifier (*Chang and Lin, 2011*) to decode the identity of the test stimuli and their match status. Stimulus identity: training sets were built by randomly picking with replacement 140 trials from the pool of test A trials (70 target A and 70 opposite-target A test stimulus presentations) and 140 trials from the pool of match B trials (70 target A and 70 opposite-target B test stimulus presentations). Testing sets were built by picking with replacement 60 trials, different from training trials, for similar conditions each. Match status: training sets were built by randomly picking with replacement 140 trials from the pool of target trials (70 target A and 70 target B) and 140 trials from the pool of non-match trials (70 opposite-target A and 70 opposite-target B). Testing sets were built by picking with replacement 60 trials, different from training trials, for similar target and opposite-target conditions. These procedures were repeated 1000 times, and classifiers were considered to perform above chance (50%) if the accuracy of the decoder was higher than chance level for more than 990/1000 iterations (p<0.01).

### Decimation approach

LIP neuronal responses were strongly influenced by both the identity (FBA) and the position (SBA) of the sample being remembered. These characteristics of neuronal responses interfered respectively with the analysis of the dynamic of identity and match selectivity and with the comparison of match selectivity between attention IN and attention OUT conditions. We equated activity between different conditions using a randomized decimation approach. This method gave qualitatively similar results as divisive normalization with respect to baseline activity (results not shown). *Equating sample A and sample B trials:* In order to test the dynamic of neuronal selectivity to the identity and to the match status of test stimuli, we used a decimation approach to equate the level of neuronal response between sample A and sample B trials based on pre-test stimulus averaged responses. For each neuron, we first computed the ratio of firing rates in a 200 ms window preceding test stimulus presentation between sample A and sample B trials. We then randomly removed from each neuron's spike trains, the number of action-potentials required to equate the pre-stimulus firing rates. For

example, if one neuron showed 20 spikes/s during the 200 ms preceding test stimuli presentation during sample A trials and 25 spikes/s during the same period during sample B trials, we randomly removed 1/5 of all the spikes emitted by this neurons during presentation of test stimuli during sample B trials. *Equating attention IN and attention OUT trials*: In order to compare selectivity to match stimuli located inside and outside each neuron's RF, a similar approach was used to decimate the spike trains of neuronal responses during attention IN to the level of attention OUT. We computed the ratio of firing rates in a 200 ms window preceding test stimulus presentation between attention IN and attention OUT. In order to base this ratio on response to visual stimuli, we excluded trials in which pre-stimulus corresponds to the delay epoch (test stimulus one in the sequence of test stimuli presentation). We then randomly removed from each neuron's spike trains as many action potentials as necessary to equate the ratio of pre-stimulus firing rate.

## Permutation test for laterality index

We used permutations methods to compare match-selectivity to test stimuli located inside or outside neurons' RFs. First, we selected randomly with replacement trials from target and opposite-target stimuli during attention IN and attention OUT conditions. We then performed ROC analyses comparing responses to target and opposite-target stimuli independently during attention IN and attention OUT conditions. This procedure was repeated 1000 times. We then computed all the combinatory differences between target/opposite-target IN and target/opposite-target OUT (1000 * 1000 direct comparisons). Levels of selectivity between attention IN and attention OUT conditions were considered significantly different if more than 95% of target/opposite-target IN minus target/opposite-target OUT measures were positive.

## Additivity

The logic of this analysis was to decompose LIP neuronal responses to target stimuli. Target stimuli were composed of the relevant color, the relevant direction and irrelevant information (such as the shape of the stimuli, the speed of movement of the dots. . ..). We therefore hypothesized that LIP neurons encoded target stimuli as follows (*Equation 1*):

$$T = C + D + I \tag{1}$$

where T is the response to target stimuli, C represents the signal related to the relevant color; D represents the signal related to relevant direction and I represents the signal to irrelevant information. However, given our task design, it was impossible to directly assess C and D. We had access to LIP neuronal responses to color-match and direction-match stimuli, which, following the same logic, can be decomposed as follows:

$$CM = C + I \tag{2}$$

$$DM = D + I \tag{3}$$

where CM and DM represent responses to color-match stimuli and direction-match stimuli respectively. Optimally, we should define color match stimuli as stimuli composed of the relevant color and the direction of the opposite-target and define direction match stimuli as the conjunction of the relevant direction with the color of the opposite-target. Unfortunately, given our task design, the low number of presentation of these stimuli did not ensure reliable signals. Therefore, in order to approximate CM, we used the conjunctions of the relevant color with any of the seven irrelevant directions. Similarly, we approximated DM using the conjunctions of the relevant direction with any of the seven irrelevant colors.

Therefore, combining *Equations 1, 2 and 3*:

$$T - I = CM - I + DM - I \tag{4}$$

Before testing our hypothesis that LIP linearly integrates color and direction-related signals, we need first to subtract signals related to irrelevant information (which in our task corresponds to responses to opposite-target stimuli) to responses to target, color-match and direction-match stimuli

signals related to irrelevant information. In this analysis, we assume that irrelevant information contained in target, color-match and direction-match stimuli are equivalent.

## Effect of feature-based attention

*Color tuning:* The slope of a linear regression fitting the neuronal response to each color quantified each neuron's color tuning. Yellow and red corresponded to values 1 and 8 of the X-axis, respectively. The amplitude of color tuning shifts was assessed by subtracting the slopes of linear regressions during sample A and sample B trials. *Direction tuning:* The preferred direction of each neuron during sample A and sample B were quantified independently by computing directional vectors defined by the following equation:

$$\begin{cases} X = \sum\limits_{i=1}^{8} FR(i) * \cos(direction(i)) \\ Y = \sum\limits_{i=1}^{8} FR(i) * \sin(direction(i)) \end{cases}$$

where *FR(i)* is the mean firing rate of the neuron to the $i^{th}$ direction (excluding match stimuli); [0 X] and [0 Y] are the Cartesian coordinates of the direction vector. The amplitude of direction tuning shifts corresponded to the angular distance between preferred direction during sample A and sample B trials. These values were normalized so that positive and negative angular distances correspond to shift toward and shift away the relevant direction, respectively.

## Data availability

Experimental data sets and analysis tools will be made available upon request from the corresponding author.

## Acknowledgements

We thank N Buerkle for animal training, the staff of the University of Chicago Animal Resources Center for expert veterinary assistance. We also thank JH Maunsell, NY Masse, K Mohan and B Peysakhovich for comments and discussion on earlier versions of this manuscript. This work was supported by NIH R01 EY019041 and NSF CAREER award 0955640. Additional support was provided by The McKnight Endowment Fund for Neuroscience.

## Additional information

### Funding

| Funder | Grant reference number | Author |
| --- | --- | --- |
| National Institutes of Health | R01 EY019041 | David J Freedman |
| National Science Foundation | 0955640 | David J Freedman |
| McKnight Endowment Fund for Neuroscience | | David J Freedman |

The funders had no role in study design, data collection and interpretation, or the decision to submit the work for publication.

### Author contributions

GI, Conceptualization, Formal analysis, Investigation, Methodology, Writing—original draft, Writing—review and editing; DJF, Supervision, Funding acquisition, Project administration, Writing—review and editing

### Author ORCIDs

Guilhem Ibos, http://orcid.org/0000-0002-9280-1280
David J Freedman, http://orcid.org/0000-0002-2485-5981

## Ethics

Animal experimentation: This study was performed in strict accordance with the recommendations in the Guide for the Care and Use of Laboratory Animals of the National Institutes of Health. All experimental procedures were in accordance with the University of Chicago Animal Care and Use Committee (IACUC), protocol #71887, of the University of Chicago and National Institutes of Health guidelines.

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
