## [Decision Letter]

Thank you for submitting your article "Sequential sensory and decision processing in posterior parietal cortex" for consideration by *eLife*. Your article has been reviewed by two peer reviewers, and the evaluation has been overseen by a Reviewing Editor and Timothy Behrens as the Senior Editor. The reviewers have opted to remain anonymous.

The reviewers have discussed the reviews with one another and the Reviewing Editor has drafted this decision to help you prepare a revised submission.

Summary:

The manuscript describes a selectivity and dynamics of LIP neurons during a task that involves matching of a conjunction of color and motion direction. The main findings reveal that one group of LIP neurons reflects stimulus identity, another represents the match and still the activity of the third group reflects the mixture of the two. The authors also present the data showing correlation between neural responses for matches and reaction times, arguing that these decision-related signals are distinct from the motor preparatory signals.

Both reviewers had a number of positive comments on the manuscript, stressing its clarity, the rich data set, and solid experimental design. However, they raised a number of issues which must be addressed before the paper can be considered for publication in *eLife*. These are listed below.

Essential revisions:

1) Provide an explanation how the data presented here go beyond what is already known about feature selectivity and category-related activity of LIP neurons reported earlier. Strengthening the link to behavior on a trial-by-trial basis could be one way to address this problem.

2) In the analysis of neural activity show the data for each monkey separately.

3) Additional analyses of the critical comparisons of identity and match selectivity should exclude 'match selective' neurons with selectivity that could be explained by motor planning."

4) Explain the advantages of the decimation approach over standard normalization for the analysis of response selectivity.

5) Address the apparent asymmetry in the analysis of identity and match selectivity, with the identity analysis involving large stimulus differences while match analysis involving all stimuli.

6) Address the comment about the use of maximum firing rate as a response metric for the reaction time correlation analysis.

7) Address the question about subtracting all non-matches in the linearity analysis.

*Reviewer #1:*

In this manuscript, Ibos and Freedman describe a study of LIP's selectivity and dynamics during a task that involves matching of a conjunction of features (color and motion direction). This work thus stands to complement prior studies of both simpler categorization tasks (e.g., direction only) and simpler match-to-sample tasks used in many classic studies. In general, the study is clearly described and the manuscript reports a rich dataset with several appropriate descriptive analyses. The general pattern of results is supported by the figures and quantifications; namely, that (a) some LIP neurons reflect stimulus identity; (b) some LIP neurons reflect the match itself, with delayed dynamics relative to identity; (c) some LIP neurons reflect a mixture; (d) a substantial proportion of these response patterns are distinct from motor preparatory signals; and (e) the match signals arise from a stronger interaction than simple additivity of the color and motion identity signals.

While I think there's much to be learned from this dataset (and generally believe that this group is furthering our understanding of what LIP might actually compute), I am struggling with the take-home from the manuscript it its current form. I am looking forward to discussion with the other reviewers on this point. The basic identity selectivity seems like a straightforward extension of several studies by this group and others, i.e., we know that LIP neurons often exhibit selectivity for relevant visual features. The match-related signals and their dynamics.… well, might these also be consistent with the category-related activity described in simpler tasks? The dynamics and distribution across neurons are interesting, and the additivity analysis is also a nice touch that points the way toward more quantitative models of what these responses might signify. Perhaps I am just spoiled by some of the excellent work by this group that has compared LIP selectivity and dynamics to, say, PFC, as studied under the same conditions. But to kick off the discussion, I'm still searching for a way to push this study a few more yards to the level of major significance.

To summarize some possible ways to enrich/expand the impact of this study, the authors could consider:

1) Errors. I realize that the task is performed at >90% accuracy, but perhaps LIP responses during errors could be leveraged to provide insight into what LIP's relation to behavior. The dissected RT correlation is nice, but given the multi-faceted selectivity and dynamics, it seems like some sort of trial-by-trial relation to behavior might be especially informative.

2) A more direct explanation of what new is learned from this study relative to both prior work on uni-dimensional categorization and in the related recent attention study by the same authors. I can't help but think I'm missing something pretty obvious about how the authors are thinking about the novelty of this study relative to what was reported in their previous attention study using this task, beyond the disambiguation of motor-related signalling.

3) Implication of a causal role. Much of the phrasing of the manuscript follows the usual convention of finding a neural correlate and assuming it plays a causal role. Although the authors have not done causal manipulations, I wonder whether this phrasing is not just conventional looseness, but rather reflects their confidence that the dynamics of these signals indeed imply a role in performing the task that is more compelling than in simpler tasks. Could this logic be laid out more clearly?

*Reviewer #2:*

The manuscript reports on neural responses in the lateral intraparietal area (LIP) of the macaque brain during a sequential visual matching task. In this task, monkeys are presented with a sample visual stimulus and up to three test stimuli. The stimuli consist of moving dots that can differ in color and direction between presentations. The sample stimulus can appear either in the response field (RF) of the recorded LIP neuron or in the opposite hemifield. The monkey must report with a manual response when the test stimulus matches the sample. All test stimuli are paired with a distractor in the opposite hemifield that provides no information about match vs. non-match.

The authors report distinct but overlapping populations of neurons in LIP with selectivity for the test stimulus identity, match vs. non-match within a neuron's RF, and match vs. non-match across both visual locations. These signals are multiplexed with residual selectivity that remains for the sample stimulus identity during presentation of the test stimuli. The authors find that selectivity for test stimulus identity generally precedes that for match vs. non-match. The authors also report a correlation between neural responses for matches and monkey's reaction times for the manual report of the match. Finally, the authors argue that identity selectivity in LIP derives from linear pooling of color and motion signals while match vs. non-match selectivity in LIP involves super linear combinations and a larger influence of feature-based attention.

The manuscript is written well and the experimental design is solid. Furthermore, the questions addressed are important and likely of interest to a large number of researchers. That said, I had numerous questions about the analyses that cause me concern. All of these questions are potentially addressable through clarification and/or further analysis. Below, I describe my questions in more detail.

1) The authors use two monkeys, but report individually on them only for one panel of the behavioral figure and for none of the neural analyses. It is important to establish which of the findings are supported independently by data from each monkey.

2) The authors have used an experimental design that allows them to find match selectivity that cannot be the result purely of motor preparation. In particular, they find neurons that respond selectivity for matches within their RF but not for matches outside their RF, even though the latter involve the same motor planning. While the authors have performed this nice analysis, I believe they have still included neurons with selectivity that may be explained by motor planning in most of their other analyses, including the timing analysis. Do the same results hold if this set of neurons is excluded?

Backing off my concern a bit, I think the authors are overly confident in attributing non-specific match selectivity to motor planning. It could be the result of motor planning, but it also could be a more abstract decision signal about matches regardless of location. Distinguishing the two would require a task with the movement but not the match decision.

3) One issue faced by the authors was disentangling multiple forms of selectivity that seem to be multiplexed by LIP responses. They dealt with this by combining ROC analyses with a "decimation" strategy that randomly removed spikes from neural responses to equate responsiveness for trials with different sample stimuli. I have multiple concerns about this. I am not sure why the decimation approach that involves randomness was needed. Why couldn't the authors use a more standard form of normalization that didn't involve randomness? Also, the authors should make clear that any approach of this variety assumes the brain knows the identity of the sample stimulus and knows how to compensate exactly for that.

Even with an alternative to decimation, the problem of multiplexing still exists. The authors have addressed it for selectivity to the sample stimulus, but I don't believe they have addressed it for mixed selectivity for test stimulus identity and match vs. non-match. This causes similar problems for a ROC-based approach. The heart of the issue is of course the mixed selectivity, so the core concern I have is that I don't think the authors have adequately acknowledged the full extent to which this poses challenges to their analyses.

4) I am worried that the comparison between identity selectivity and match selectivity may be biased. In particular, the description in the Methods seems to suggest identity selectivity is assessed comparing test stimulus A to test stimulus B and not using any intermediate stimuli, while match selectivity involves all stimuli, including ones very close to the match. If this is the case, it would be problematic because the former is an easier discrimination than the latter and it would call into question the comparisons between the two. Perhaps I am misunderstanding this, though, and the authors restricted the match selectivity analysis to the same stimuli as the identity selective analysis. In that case, I withdraw this concern.

5) I have a number of questions for the reaction time correlation analysis. I worry about the robustness of using the maximum firing rate as the metric for the neural responses. This treats very small maximum rates the same as large ones. It also discounts that the maximum may be achieved after the decision, sometimes even after the manual response. It would be nice to know that this analysis is robust to these concerns. Are similar results obtained using a metric that involves exceeding a threshold response level (recognizing this approach will involve some trials not being able to be included)? To what extent is the correlation driven by the ceiling caused by the actual movement? (For example, the absence of data points in the bottom right of the middle panel of Figure 5 seems to be driving the correlation.) Are the latencies (intercept of the correlation) reasonable given the authors conclusions?

6) For the linearity analysis, I didn't understand why the authors would subtract all non-matches. This treats very close non-matches as identical to very far non-matches. More generally, it seems odd to start an analysis about linearity with an extremely non-linear first step in the analysis.

---

## [Author Response]

*Essential revisions:*

*1) Provide an explanation how the data presented here go beyond what is already known about feature selectivity and category-related activity of LIP neurons reported earlier. Strengthening the link to behavior on a trial-by-trial basis could be one way to address this problem.*

We would like to thank reviewer 1 for encouraging us to clarify how the conclusions of the current study differ from the conclusions of our previous studies. We hope that they agree that the following explanations and the changes made to our manuscript better portray the significance of our study. First, reviewer 1 suggested that we analyze error trials in order to link neuronal activity to behavioral responses. Such data would be of great interest and we tried to perform these analyses. Unfortunately, the number of error trials, which was very low for each condition, did not yield enough statistical power to draw reliable conclusions about these results nor did it allow us to detect trends in how erroneous behaviors affect LIP neuronal responses. However, we have improved the analysis of the correlation between neuronal dynamics and monkeys’ behaviors (see point 6).

Before responding to reviewer 1’s individual points, we would like to emphasize the novelty of our study compared to our previous work and that from other groups. To our knowledge, our study is the first:

To characterize overlapping populations of identity and match selective neurons in LIP (described in our study as a continuum). Previous studies either showed selectivity to relevant stimuli (in what is usually called a priority or salience map), or to non-spatial aspects of stimuli (such as individual features or category selectivity). These two types of information were not compared in previous studies.To show a clear temporal relationship between identity and match selectivity (identity preceding match selectivity) in LIP.To link match selectivity in LIP to monkeys’ behavioral choices.To propose that LIP represents the identity of task-relevant stimuli (which in our study are composed of multiple visual features) by reading out and combining semi-linearly (see point 7 of this reviews) signals from upstream cortical areas.To propose that LIP match selectivity reflects computations related to the comparison of bottom-up sensory and top-down cognitive signals.

In the Discussion, we discuss all of these results within a new and coherent theoretical framework for how LIP uses sensory and cognitive information to mediate the decision-making process. Specifically, we propose that LIP is an interface between different sources of bottom-up and top-down signals, integrating task relevant signals in order to compute decision variables.

More specifically, we would like to address reviewer 1’s specific points and discuss how the pattern of results described in our study builds on, goes beyond, and differs from the discoveries from our previous studies. We understand reviewer 1’s concern came from 2 specific points:

A) The identity selectivity is not surprising given that “LIP neurons often exhibit selectivity for relevant visual features”.

This is an important point, which we have addressed and clarified in the revised manuscript. Our two previous articles (Ibos & Freedman 2014, 2016) showed that, depending on feature-based or space-based attention, color and direction tuning of LIP neurons were shifted toward the relevant features. While LIP neurons were indeed selective to task relevant features (color and direction), they were not necessarily preferentially tuned to the color or the direction which was relevant on each trial. For example, the direction tuning shifts were subtle and relatively few neurons were tuned to direction A during sample A trials and to direction B during sample B trials. Therefore, it was difficult to predict how LIP neurons encode conjunctions of relevant color and relevant motion direction in order to facilitate the monkeys’ decisions. The current study describes this aspect of our data which was not described in our previous reports.

Moreover, we would like to stress that, in our 2 previous studies, we excluded from our analyses of feature selectivity neuronal activity acquired during presentation of match stimuli, and analyzed selectivity to either the color or the direction of test stimuli. However, we never described how LIP responded to conjunctions of color and direction. Therefore, the previous studies accounted for how LIP encoded color or direction features, it did not account for how LIP neurons encoded their conjunction. Nothing in these previous reports described how LIP combined these two sources of sensory information (color and motion direction), nor addressed mechanisms by which LIP might encode either the match status of the stimuli or their identity. Testing how LIP neurons encode conjunctions of relevant features is complementary but also fundamentally different than asking how attention affects LIP feature representations. We were able to address this distinction because of our task design: monkeys had to base their decision on multiple features (color and direction), each encoded in visual cortex by different pools of neurons located in different brain areas (presumably V4 for color, MT for motion-direction). Since monkeys were not able to base their decisions on a single feature (as it was the case in most, if not all, studies of decision-making in LIP), neurons involved in the match/non-match decision needed to combine both sources of information.

B) “The match-related signals and their dynamics […] might […] also be consistent with the category-related activity described in simpler tasks.”

We realize now that we need to make an additional effort in considering the relationship between this work and previous results from our lab during visual categorization tasks. The two lines of work differ in important ways.

It is true that some of our previous work about category selectivity in LIP described some aspect of the neuronal response to match stimuli. Supplementary Figure 5 of (Freedman & Assad, 2006) shows that category selectivity in LIP is highly similar during sample and test presentation. For example, a neuron selective to category 1 during the sample and delay period was also selective to category 1 during test presentation, whether the presented stimulus was a match or a non-match. Therefore, this selectivity of LIP neurons to abstract category membership of test stimuli is related to encoding the task-relevant features of test stimuli, but cannot account for LIP match selectivity described in the current study.

Another study from our group (Swaminathan, Masse & Freedman, 2013) compared responses of LIP and MIP neurons to matching and non-matching test stimuli. This study described a significant modulation of LIP neuronal response by the match status of the test stimuli (according to a two-way ANOVA) but specifically stated that “LIP response was more dependent on both the match/non-match status as well as category of test stimuli”. In addition, that study showed that the LIP population response to test stimuli was similar during match and non-match trials (Figure 2 of (Swaminathan et al., 2013)). We agree with reviewer 1 when she says that our description of match selectivity is consistent with our previous work. However, the current study goes well beyond previous work to give insight into how task-relevant feature encoding relates to, and aids in the computation of, decision variables.

In our opinion, and given the points mentioned above, the current study represents an important step forward in the understanding of selectivity to sensory and cognitive factors in LIP. In order to clarify these points, we made several changes in the manuscript:

Introduction: “Our previous studies and the model framework which accompanied them (Ibos and Freedman, 2014, 2016) can potentially account for the encoding of individual spatial and non-spatial features in LIP. However, it did not address the role of LIP in solving tasks in which decisions rely on grouping different sensory feature representations and comparing them to an internal cognitive model of task-relevant information.”

Introduction: “LIP has been proposed to encode the behavioral salience (Arcizet et al., 2011; Bisley and Goldberg, 2003, 2010; Gottlieb et al., 1998; Ipata et al., 2009; Leathers and Olson, 2012) of stimuli, to transform sensory evidence into decisions about the target of saccadic eye movements (Gold and Shadlen, 2007; Huk and Shadlen, 2005; Leon and Shadlen, 1998; Shadlen and Newsome, 2001) and to encode cognitive signals such as rules (Stoet and Snyder, 2004) or abstract categories (Freedman and Assad, 2006; Sarma et al., 2015; Swaminathan and Freedman, 2012) independently of LIP’s role in spatial processing(Rishel et al., 2013). The present study focuses on understanding how LIP jointly encodes sensory, cognitive and decision-related information during a complex memory-based visual-discrimination task in which decisions rely on comparing the identity of observed stimuli to the identity of a remembered stimulus.”

Discussion, first paragraph: “We found that, along the visuo-decision continuum, the encoding of the relevant conjunctions switched from additive to super-additive. […] We propose that LIP is composed of a visuo-decision continuum of neurons involved to different degrees in 1) the integration of bottom-up sensory inputs, 2) the linear grouping of these disparate sensory signals and 3) their transformation into a signal related to monkeys’ decisions about the relevance of the visual stimuli.”

Discussion, paragraph called “Toward a new model of decision-making in LIP?”: “The super-additivity observed for match-selective neurons in LIP hypothetically is the outcome of a computation related to the comparison of the identity of the stimuli with top-down signals, presumably originating in PFC (Mante et al., 2013), related to which features are behaviorally relevant. The selectivity for sample identity across each trial epoch (Figure 7) could reflect the integration of top-down signals related to the identity of the relevant stimuli kept in working memory.”

*2) In the analysis of neural activity show the data for each monkey separately.*

As proposed by reviewer 2, we now show the data separately for each monkey. Figure 4—figure supplement 1 shows the time course of identity selectivity, match selectivity and of the visuo-decision index for each monkey and task. In addition, in each scatter plot, data from monkey M are shown with full black dots, data from monkey N are shown with empty circles.

During the two-location DCM task, 27/36 and 19/36 of Monkeys N’s neurons were selective to the identity and match status of the test stimuli, respectively. Similarly, 18/24 and 22/24 of monkey M’s neurons were selective to the identity and match status of the test stimuli, respectively.

During the one-location DCM task, 24/31 and 15/31 of monkey N’s neurons were selective to the identity and the match status of the test stimuli, respectively. Similarly, 23/56 and 47/56 of monkey M’s neurons were selective to the identity and the match status of the test stimuli, respectively.

This shows that both types of information are encoded similarly in both monkeys. However, an analysis of the visuo-decision indices showed asymmetric distributions. Monkey N’s VDI were shifted toward positive values (identity encoding), while monkey M’s VDI were shifted toward negative values (match status encoding). However, despite this asymmetry, both types of information are encoded in each monkey and the main conclusions of our analyses are marginally impacted by separating data from each monkey. Below, we provide a detailed description of the few analyses and results which were impacted:

Correlation between neuronal response and reaction times: Removing bilaterally selective neurons: this control analysis is impacted for monkey N if we remove neurons with bilateral selectivity to match stimuli (corr coef=0.01; p=0.9). This loss of statistical power for this control can be explained by the decrease in sample size. It is now explained in the Results section.Correlation between additivity index and VDI: full datasets: Additivity index and VDI of monkey N’s neurons are not significantly correlated (corr coef=-0.20, p=0.09) but follow the same trend as the ones of monkey M. Removing bilaterally selective neurons: results are statistically impacted for both monkeysbut follow the same trends (monkey M: corr coef=-0.37, 0.17; monkey N: corr coef=-0.23, p=0.26). This lack of statistical power can also be attributed to smaller sample sizes (monkey M, N=15; monkey N=25). This has been explained in the relevant section of the Results section.Correlation between amplitude of direction tuning shifts and VDI: full datasets: resultsare impacted for monkey N (corr coef=-0.08, p=0.53). Removing bilaterally selective neurons: results are impacted for both monkeys (corr coef=-0.06, p=0.68). This is not surprising given that direction tuning shifts are more subtle than color tuning shifts and therefore more difficult to characterize (Ibos & Freedman 2014, 2016). Decreasing sample’s size strongly reduce the power of our tests.

Given the small impact of separating data from each monkey from a statistical point of view, and for sake of clarity of our manuscript, we now plot each monkey’s data separately but present statistics for both monkeys grouped together. We clarify this point now at the beginning of the Results section: “We recorded the activity of 201 individual LIP neurons while monkeys performed the tasks (74 during two-location and 127 during one-location DCM tasks). […] However, unless mentioned otherwise, analyzing data separately for each monkey did not affect the outcomes of the following analyses.”

*3) Additional analyses of the critical comparisons of identity and match selectivity should exclude 'match selective' neurons with selectivity that could be explained by motor planning."*

Our analyses included neurons selective to either the identity of the stimuli or to their match status. We didn’t differentiate between neurons selective to match stimuli located only inside neurons’ RF (N=21, unilaterally match selective neurons) and neurons selective to match stimuli located both inside and outside neurons’ RF (bilaterally match selective neurons, N=20). This later population could reflect motor-related signals which are by definition linked to monkeys’ behaviors. It is therefore crucial to control for the possibility that the effects related to match selectivity described in our study are not due to this very specific population of bilaterally match selective neurons. We have now included controls for this issue in supplementary figures.

We focus our control analysis on the population of neurons acquired during the two-location DCM task (N=60/74). As a control, we removed from this pool of neurons the population of 20 neurons which showed bilateral selectivity and show that it does not affect our conclusions. LIP still shows a continuum of mixed selectivity from purely identity-selective to purely match-selective (Figure 4—figure supplement 2). Moreover, removing neurons most closely associated with motor-related selectivity (the neuronal population bilaterally selective for the match status of test stimuli) does not change the sequential timing of identity and match selectivity. Finally, we still observe the inverse correlation between the visuo-decision index and the choice probability (correlation coefficient between firing rates dynamics and monkeys’ reaction times, this control has been added as Figure 5—figure supplement 1A).

However, we had to make a choice: should we remove this population of 20 bilaterally match selective neurons from our pool or should we include them? The response to this problem was not trivial. It is directly related to reviewer 2’s comment in which they asked us to temper our claim that bilateral match selectivity reflects a motor-preparation signal. We completely agree with reviewer 2 about this point. Our task was not designed to differentiate motor/reward/abstract and non-spatial decision processes, which is an issue we are addressing with ongoing and planned follow-up studies.

Given the minor impact of removing bilaterally match selective neurons (shown in supplementary figures) on our results, we decided to group bilateral and unilateral match selective neurons together, and to include these data in our results, but also to present these controls in supplementary figures. This allowed us to merge data from both datasets (two-location and one-location delayed conjunction matching tasks, Ibos & Freedman 2014, 2016).

In addition to what we originally claimed in the Results section – “However, our task was not specifically designed to characterize the exact nature of non-spatial match-selectivity signals, which could reflect different processes (e.g. motor-preparation, motor-execution or even reward expectancy)” – we have included additional details about this issue in the revised manuscript. First, we have removed the following sentence from our manuscript: “In the following, unilateral match selectivity will be associated with the decision process while bilateral selectivity will be linked to motor-planning.” Second, we have modified the end of the paragraph called “Typical LIP neuronal response” (in which we describe responses of individual LIP neurons). We now no longer directly refer to bilateral match selectivity as related to motor-preparation, we instead use the more neutral terminology of bilateral match selectivity (Figure 3; along the text):

“Moreover, selectivity for the identity and match status of test stimuli were not mutually exclusive and some neurons multiplexed these signals. […] In the following, we will characterize these two types of selectivity (identity, match status) at the population level along with the influence of sample identity on the neuronal response.”

*4) Explain the advantages of the decimation approach over standard normalization for the analysis of response selectivity.*

In our study, we needed to equate level of neuronal activity between either sample A and sample B trials, or attention IN and attention OUT conditions in order to compare dynamics of identity and match selectivity. We used a randomized decimation approach. We could have used a more standard normalization method in which the firing rates (average over a period of time) would have been divided by a factor representing the ratio of response between sample A and B or attention IN and OUT conditions prior to stimulus presentation. The two methods are nearly equivalent. However, using a decimation approach allowed us to work with (decimated) spike trains instead of averaged firing rates and therefore to keep intact the dynamic of the neuronal response. Our method is therefore more conservative of the dynamics of neuronal responses on a trial to trial basis. This point has been clarified in the manuscript:

“Therefore, prior to analyzing data, we equated each neuron’s responses during either sample A or B trials (depending on each neuron’s preference) using a randomized decimation approach (an approach more conservative of spike train dynamics than standard normalization methods) based on the ratio of firing rate between sample A and sample B trials prior to test stimulus onset (see Methods).”

In the second part of their remark, reviewer 2 asks us to acknowledge two kinds of challenges raised by mixed selectivity in LIP: A) a conceptual challenge related to how the brain areas read out LIP population activity showing mixed selectivity, and B) a technical challenge related to our analysis.

A) We acknowledge the computation challenge it represents for the areas reading out LIP activity. We have now included a sentence in the last paragraph of the Discussion: “The observation of mixed selectivity poses a decoding challenge for downstream areas in reading-out the activity of LIP neurons.”

B) It is true that mixed selectivity for sample identity and test stimulus identity posed a challenge to our analyses. For example, prior to decimation, a neuron selective to the identity of the sample during test presentation would exhibit an artefactual identity selectivity in our analyses. However, the mixed selectivity of match and identity information did not pose such a challenge in our analyses, since they are symmetrical (we use the exact same neuronal responses to characterize identity and match selectivity). Therefore, there is no reason why identity selectivity would interfere with match selectivity (and vice versa). For example, a stronger response to test stimulus A during sample A and B trials compared to test stimulus B during sample A and B trials will show strong identity selectivity but weak match selectivity according to our ROC analyses.

*5) Address the apparent asymmetry in the analysis of identity and match selectivity, with the identity analysis involving large stimulus differences while match analysis involving all stimuli.*

We realize that our original description of this analysis approach was confusing, which we have clarified in the revised manuscript. The following response also stands for point 7. When we talked about non-match stimuli in these comparisons and all along the study, we referred to either test stimulus A (during sample B trials) or test stimulus B (during sample A trials) when they did not match the sample stimulus. Therefore, the ROC analyses characterizing identity and match selectivity were symmetrical. This has been clarified in the revised manuscript. The terms “match stimuli” and “non-match stimuli”have been replaced by “target” and “opposite-target stimuli”: “test stimuli (presented at the sample location) were randomly picked among 3 types of stimuli: (1) target stimuli, matching the sample, (Connor et al., 1997) opposite-target stimuli (i.e. the sample stimulus which was not presented during that trial, e.g. test stimulus A during sample B trials), (McAdams and Maunsell, 2000) any of the remaining conjunctions of color and motion-direction.”

*6) Address the comment about the use of maximum firing rate as a response metric for the reaction time correlation analysis.*

In this analysis, we wanted to link the dynamics of neuronal responses to the monkeys’ behavioral response. In the original version of this manuscript, we used maximum firing rates as a measure of neuronal dynamic. As pointed out by reviewers, this method is unconventional and our additional analyses show that it is not a crucial approach for this point. We have replaced it by a more standard method. Now, we first z-score the time course of each trials’ firing rates (after convolving each trial’s spike trains with a Gaussian (σ=15°)). We then correlate each trial’s z-scores with each trial’s manual reaction time (this method is similar to the one described and used by several studies, e.g. Cook & Maunsell, 2002). This method allowed us to address the issue raised by reviewers about the previous version of the analysis. Results are now shown in Figure 5. We have also modified the paragraph describing these results in the Results section:

“To test this hypothesis, we analyzed whether the time course of each neuron’s activity was correlated to the monkeys’ RTs. […] Artefactual correlation in this kind of analyses can emerge when monkeys’ detection rates fluctuate with the timing at which the stimuli were presented during each trial (Kang and Maunsell, 2012), leading to more presentation of the target stimulus at a certain trial epoch (e.g. larger number of target detected during the presentation of the 1^st^ test stimulus).”

*7) Address the question about subtracting all non-matches in the linearity analysis.*

We tried to understand how LIP neurons combine different sources of sensory information about the color and the motion direction of test stimuli. Our approach involved a subtracting normalization method which, we realize now, needs to be clarified. Reviewers’ remarks referred to two main concerns:

For the linearity analysis, reviewer 2 asked about the analysis step of subtracting neuronal activity for non-match trials. This concern relates to point #5, and our initially confusing explanation of the term “non-match” in the original manuscript. This has been clarified in the text as we replaced the term “non-match” by “opposite-target”.

Reviewer 2 raised a crucial point: “it seems odd to start an analysis about linearity with an extremely non-linear first step in the analysis.”

The logic of this analysis was to decompose LIP neuronal responses to target stimuli. Target stimuli were composed of the relevant color, the relevant direction and irrelevant information (such as the shape of the stimuli, the speed of movement of the dots….). We therefore hypothesized that LIP neurons encoded target stimuli as follow (equation 1):(1)T=C+D+I

Where T is the response to target stimuli, C represents the signal related to the relevant color; D represents the signal related to relevant direction and I represents the signal to irrelevant information. However, given our task design, it was impossible to directly assess C and D. However, we had access to LIP neuronal responses to color-match and direction-match stimuli, which, following the same logic, can be decomposed as follow:(2)CM=C+I(3)DM=D+I

Where CM and DM represent responses to color-match stimuli and direction-match stimuli respectively. Optimally, we should define color match stimuli as stimuli composed of the relevant color and the direction of the opposite-target stimulus and define direction match stimuli as the conjunction of the relevant direction with the color of the opposite target target-stimulus. Unfortunately, given our task design, the low number of presentation of these stimuli didn’t ensure reliable signals. Therefore, in order to approximate CM, we used the conjunctions of the relevant color with any of the 7 irrelevant directions. Similarly, we approximated DM using the conjunctions of the relevant direction with any of the 7 irrelevant colors. Therefore, combining equations 1, 2 and 3:(1)T−I=CM−I+DM−I

Before testing our hypothesis that LIP linearly integrates color and direction related signals, we needed first to subtract signals related to irrelevant information (which in our task correspond to responses to opposite-target stimuli) to responses to target, color-match and direction-match stimuli. In this analysis, we assume that irrelevant information contained in target, color-match and direction-match stimuli are equivalent.

This demonstration has been added in the Methods of the manuscript.

“Then, we calculated each neuron’s *additivity-index (Additivity-index=*match – (color + direction)). Despite its name, this index does not reflect purely linear processes since it resulted from subtractive normalization (a non-linear computation) in order to take into account relevant information only. An *additivity-index* of 0 corresponds to a pseudo-linear model in which neurons encode conjunctions of features by summing feature-specific sensory inputs.”